# 1,2-Amino oxygenation of alkenes with hydrogen evolution reaction

Shengzhang Liu[1,2,4], Shengchun Wang[3,4], Pengjie Wang[3], Zhiliang Huang[3] ✉,
Tao Wang[1] ✉ & Aiwen Lei ●[1,3] ✉

1,2-Amino oxygenation of alkenes has emerged as one of the most straight-forward synthetic methods to produce β-amino alcohols, which are important organic building blocks. Thus, a practical synthetic strategy for 1,2-amino oxygenation is highly desirable. Here, we reported an electro-oxidative inter-molecular 1,2-amino oxygenation of alkenes with hydrogen evolution, removing the requirement of extra-oxidant. Using commercial oxygen and nitrogen sources as starting materials, this method provides a cheap, scalable, and efficient route to a set of valuable β-amino alcohol derivatives. Moreover, the merit of this protocol has been exhibited by its broad substrate scope and good application in continuous-flow reactors. Furthermore, this method can be extended to other amino-functionalization of alkenes, thereby showing the potential to inspire advances in applications of electro-induced N-centered radicals (NCRs).

As one type of basic frame with unique physiological activity, β-amino alcohol motifs widely exist in pharmaceuticals[1], natural products[2], and ligands[3] (Fig. 1a). Due to their significant importance, the synthesis and transformation of β-amino alcohols have drawn much attention from synthetic chemists and pharmacists[2,4,5]. Forming β-amino alcohols in a single-step, 1,2-amino oxygenation of alkenes represents one of the ideal routes toward this synthetic goal. Over the last decades, 1,2-amino oxygenation has achieved several breakthroughs[6-9]. Since 1975, transition-metal (such as Pd, Os, Rh, Cu, Fe, Mn, Ir, etc.) catalyzed alkenes 1,2-amino oxygenation has undergone a flourishing development[7,10-19]. Despite their excellent regioselectivity, expensive transition-metal catalysts and/or complex ligands have propelled the development of alternative approaches. To avoid the use of metal-catalyst, oxidation-induced strategy has recently served as another fascinating way to synthesize β-amino alcohols with sacrificial oxidants (including hypervalent iodines[9,20,21], peroxides[22], diazodicarboxylates[23], fluor-containing oxidants[24,25], TEMPO[26,27], etc.). While these methods ensure efficient approaches, their compatibility with oxygen and nitrogen sources may be problematic. As green and powerful tools, photo-induced organic

transformations have been attractive synthetic methods to produce amino alcohol derivatives[28-32]. With a unique property for sustainable and practical synthetic methods, electro-induced alkenes transformation involving N-centered radicals (NCRs) has attracted extensive attention from chemists[33-35]. A representative report is an azidooxygenation of alkenes via electro-oxidation, in which TEMPO and $N_3^-$ were applied as oxygen and nitrogen sources, respectively[36].

Although numerous attractive methods have been developed for this ideal synthesis, several challenges still remain in the 1,2-amino oxygenation approach (Fig. 1b)[6]. One internal challenge is the precise control of regio- and chemoselectivity in this transformation. Since the reacting oxygen and nitrogen nucleophiles generally showed similar reactivity, other potential reactions, including diamination, 1,2-oxyamination, and dioxygenation, might adversely affect the desired 1,2-amino oxygenation. Another challenge in the application is the development of mild methods with good compatibility for both oxygen and nitrogen sources, especially for O–H and N–H compounds that represent a direct and atom-economical route. Thus, developing an efficient, cheap, and easy-to-handle method to

[1]National Research Center for Carbohydrate Synthesis and Jiangxi Province Key Laboratory of Chemical Biology, Jiangxi Normal University, Nanchang 330022 Jiangxi, P. R. China. [2]College of Traditional Chinese Medicine, Jiangxi University of Chinese Medicine, Nanchang 330022 Jiangxi, P. R. China. [3]The Institute for Advanced Studies (IAS) and College of Chemistry and Molecular Sciences, Wuhan University, Wuhan 430072 Hubei, P. R. China. [4]These authors contributed equally: Shengzhang Liu, Shengchun Wang. ✉e-mail: zlhhx@live.cn; wangtao@jxnu.edu.cn; aiwenlei@whu.edu.cn

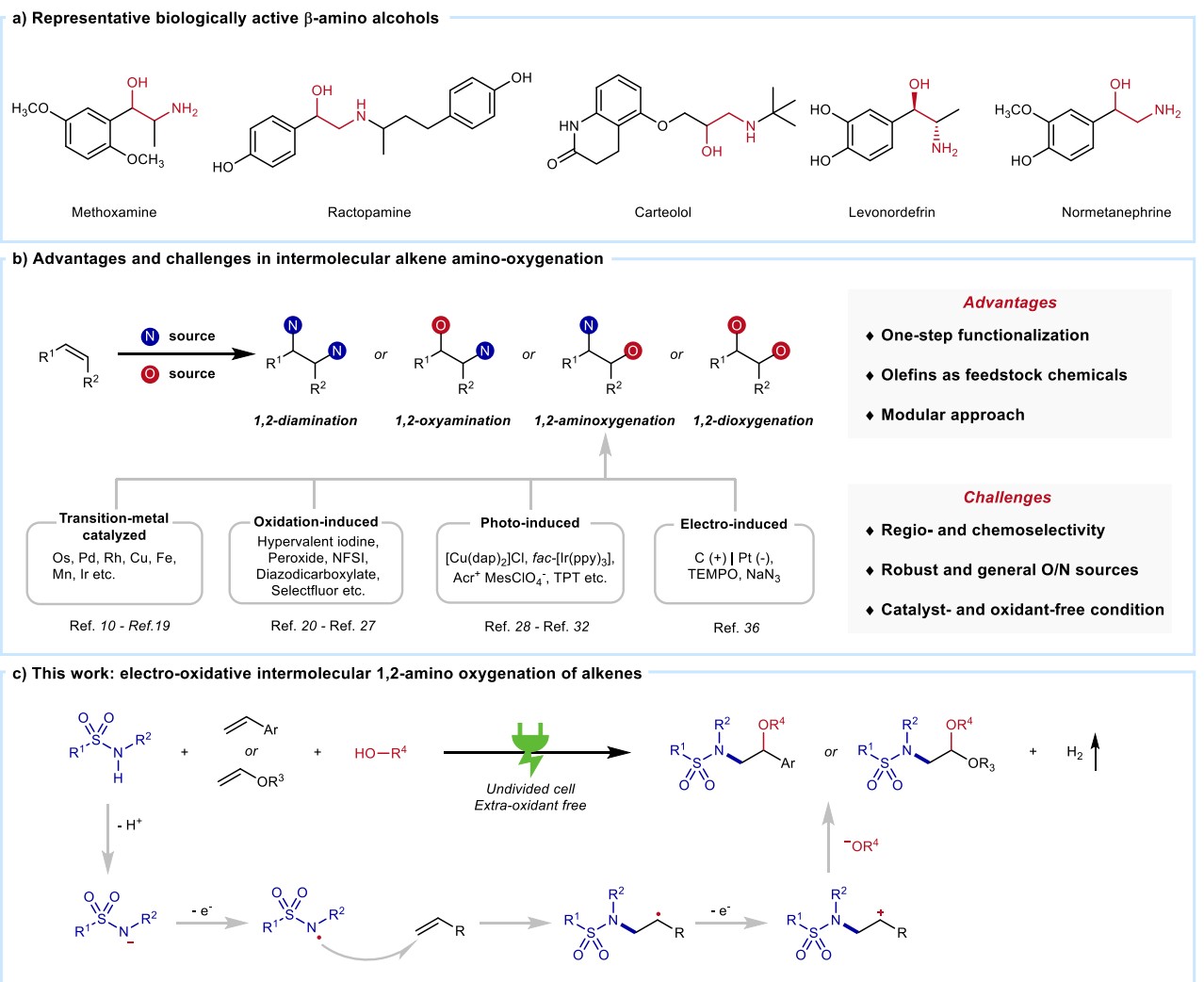

**Fig. 1 | Introduction. a** Representative biologically active β-amino alcohols. **b** Advantages and challenges in intermolecular alkenes 1,2-amino oxygenation. **c** Outline of this work: electro-oxidative 1,2-amino oxygenation of alkenes.

accomplish alkene 1,2-amino oxygenation with simple O–H and N–H functionalities are desirable, yet challenging.

Based on our previous works on electro-induced NCR chemistry[37–39], we conceived a feasible electro-oxidative 1,2-amino oxygenation of alkenes to address these challenges (Fig. 1c). As the initiation of this hypothesis, an *anti*-Markovnikov addition between NCRs and alkenes can provide high regioselectivity for such organic transformation. Moreover, the hydrogen evolution in the cathode can increase the concentration of alkoxy anions (⁻OR), which may further promote the formation of C–O bonds. Meanwhile, the direct utilization of O–H and N–H compounds may ensure the wide applicability of this synthetic method. Therefore, we report here an efficient, transition-metal-catalyst- and oxidant-free alkenes 1,2-amino oxygenation via an electro-induced NCRs pathway, providing a practical route toward synthesizing β-amino alcohol derivatives.

## Results
### Condition optimization for electro-oxidative 1,2-amino oxygenation
Usually, solvents affect the lifetime and reactivity of NCRs. Therefore, various solvents involving O–H functionalities were initially examined with *N*-methyl-*p*-toluenesulfonamide **1a** and styrene **2aa** as the model substrates. Unfortunately, the desired 1,2-amino oxygenation products were not detected in AcOH, EtOH, 2,2,2-trifluoroethanol (TFE), and

hexafluoroisopropanol (HFIP) (see Supplementary Table 1). Though the 1,2-amino oxygenation was successfully achieved in MeOH, those tested reactions were limited in their moderate yields (see details in Supplementary Table 2). Combined with our previous works, the mixed solvents were attempted, including dichloromethane (DCM)/TFE, MeCN/TFE, and DCM/HFIP. When the mixture of DCM (4 mL) and TFE **3** (2 mL) was used, the 1,2-amino oxygenation was successfully accomplished to form target product **4aa** in 76% isolated yield by using 2 equivalents 1,8-diazabicyclo[5.4.0]undec-7-ene (DBU) as the base (Table 1, entry 1). DBU is essential for the high yield, as the low conversion of **1aa** and no target product detected in its absence (entry 2), while lower yields were obtained with the use of K₃PO₄ or Cs₂CO₃ instead of DBU (entries 3 and 4). The separated yield of **4aa** slightly reduced when the electron current was decreased to 2 mA or increased to 10 mA, even with the same electron quantity (entries 5 and 6). The utilization of carbon rod or Ni plate as cathode could also furnish the product **4aa**, but in lower yields (entries 7 and 8). This electro-chemical conversion was compatible with air in which the products were formed in 60% yield (entry 9). The control experiment showed that the current is necessary for this transformation (entry 10).

### Scope of substrates
With the optimized protocol for alkene 1,2-amino oxygenation, we examined the scope of alkenes using *N*-methyl-*p*-toluenesulfonamide

## Table 1 | Optimization of the electro-oxidative 1,2-amino oxygenation of alkenes

| Entry | Variations | Yield (%)[a] |
|---|---|---|
| 1 | No | 76 |
| 2 | Without DBU | n.d. |
| 3 | $K_3PO_4$ instead of DBU | 55 |
| 4 | $Cs_2CO_3$ instead of DBU | 38 |
| 5 | 2 mA, 10 h | 74 |
| 6 | 10 mA, 2.5 h | 70 |
| 7 | Carbon rod as cathode | 66 |
| 8 | Ni plate as cathode | 67 |
| 9 | Under air | 60 |
| 10 | Without current | n.d. |

Standard conditions: graphite rod anode (Φ 6 mm), Pt plate cathode (15 mm × 15 mm × 0.3 mm), constant current = 4 mA, **1a** (0.30 mmol), **2aa** (0.90 mmol), DBU (0.60 mmol), TBABF$_4$ (0.30 mmol), DCM/TFE (4 mL/2 mL), r.t., 5 h, undivided cell, and nitrogen.
*n.d.* not detected.
[a]Isolated yields.

**Fig. 2 | Scope of substrates. a** Scope of alkenes. Reaction conditions: graphite rod anode (Φ 6 mm), Pt plate cathode (15 mm × 15 mm × 0.3 mm), constant current = 4 mA, sulfonamides (0.3 mmol, 1 equiv.), alkenes (0.9 mmol, 3 equiv.), DBU (0.6 mmol), TBABF$_4$ (0.3 mmol), DCM/TFE (4/2 mL), r.t., 5 h, undivided cell under N$_2$. **b** Scope of sulfonamides. [c]Alkenes (1.8 mmol, 6 equiv.) were used. [d]Constant current = 5 mA, 6 h, DBU (0.39 mmol, 1.3 equiv.). [e]Alkenes (3.0 mmol, 10 equiv.) were used. All yields are isolated yields.

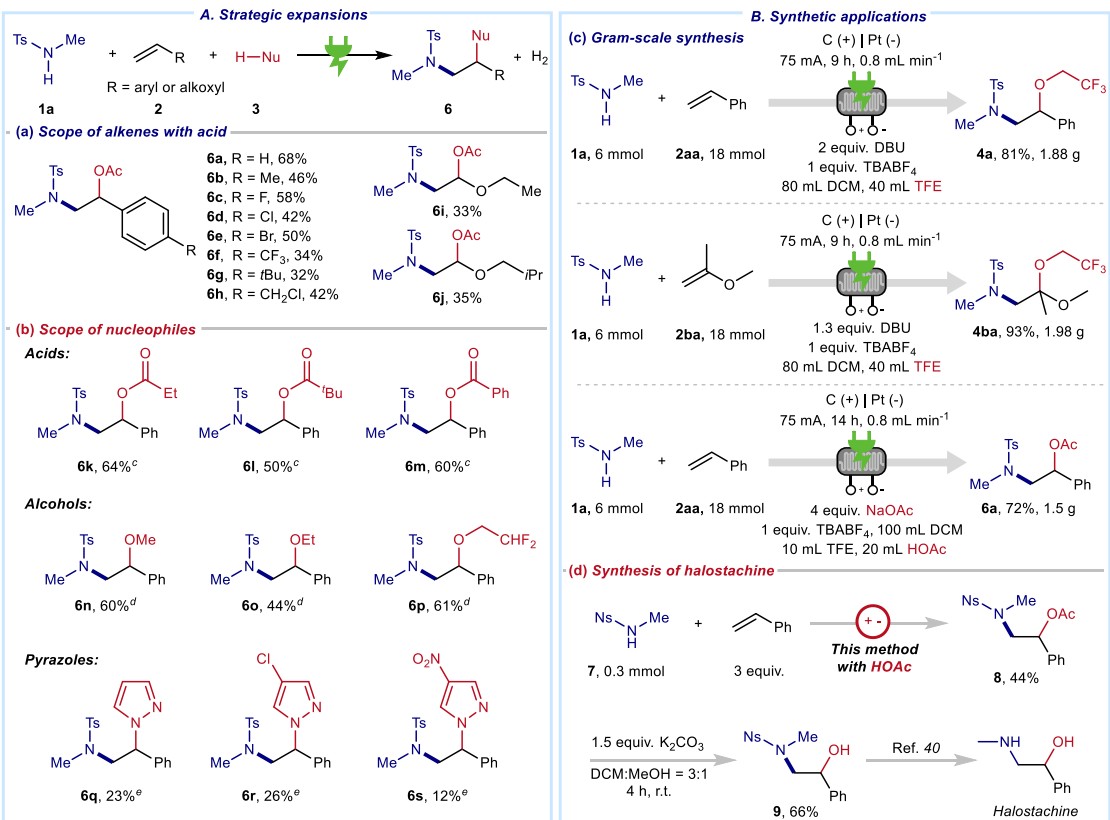

**Fig. 3 | Strategic expansions and synthetic applications. A** Strategic expansions. **a** Scope of alkenes with acids. [a]Reaction conditions: **1a** (0.3 mmol), alkenes (1.8 mmol), acetic acid 36% aqueous solution (AR, HOAc) (1.0 mL), TBAOAc (0.3 mmol), NaOAc (1.2 mmol) as base, in DCM/TFE (5 mL/0.5 mL), C anode, Pt cathode, undivided cell, 6 mA, room temperature, N₂, 5.0 h, isolated yield. **b.** Scope of nucleophiles. [c]Reaction condition: TBABF₄ (0.3 mmol), acids (1.8 mmol) and sodium carboxylate (1.2 mmol) were used instead of acetic acid 36% aqueous solution (AR, HOAc) (1.0 mL) and NaOAc. [d]Reaction condition: **1a** (0.3 mmol), **2a** (0.9 mmol), DBU (0.6 mmol), TBABF₄ (0.3 mmol), DCM/ROH (2 mL/4 mL). [e]Reaction condition: **1a** (0.3 mmol), **2a** (1.8 mmol), DBU (0.6 mmol), TBABF₄ (0.3 mmol), pyrazoles (1.8 mmol) in DCM/TFE (5 mL/0.5 mL). **B** Synthetic applications. **c** Gram-scale synthesis in continuous-flow electro-reactor. **d** The synthesis of halostachine.

**1aa** in the mixture of DCM and TFE (Fig. 2a). Various mono-functionalized styrenes with *ortho*-, *meta*- or *para*-substituents were suitable radical acceptors to afford the corresponding products in moderate to good yields (**4aa–4am**). Notably, though alkenes with electron-withdrawing groups transformed to the target products smoothly (**4an** and **4ao**), this 1,2-amino oxygenation was completed in low yields using alkenes with strong electron-donating groups (**4ap** and **4aq**) or naphthalene (**4ar**). In addition, alkene with a sensitive functional group, for example, benzyl chloride, was well tolerated in this condition (**4as**). Moreover, alkenyl ether derivatives were also compatible with this electro-oxidative 1,2-amino oxygenation. Various alkenyl ethers were smoothly transformed to target products in moderate yields (**4at–4ba**). In addition, unactivated 1,1-disubstituted alkene **2bb** successfully realized amino oxygenation to afford products **4bb**.

Subsequently, further exploration of sulfonamides was carried out under electro-oxidative conditions (Fig. 2b). A series of functionalized benzenesulfonamides with electron-donating or electron-withdrawing groups were satisfactory amination reagents (**5a–5j**). Moreover, sulfonamides with (hetero)cyclic motifs were also suitable for producing products in moderate yields (**5k–5m**). Furthermore, other *N*-alkyl substituted sulfonamides well performed in this 1,2-amino oxygenation (**5n–5r**). With the respect to limitation, secondary alkylamines and primary amides failed toward the target products under the optimized conditions. When the activated methylene was present on the nitrogen atom, the amino oxygenation was also achieved in low yields (**5s, 5t**). More efforts to address these limitations are currently going on.

## Strategic expansions and synthetic applications

Next, a series of experiments were performed to demonstrate the expansion of this strategy (Fig. 3A). Additional scope of alkenes with HOAc illustrated that this method could be well expanded to other acids with good functional group compatibility (Fig. 3A-a, **6a–6j**). In addition, this electro-induced alkenes di-functionalizations were successfully achieved with other nucleophiles, including acids, alcohols, and even pyrazoles (Fig. 3A-b, **6k–6s**).

To show the potential in further application, this synthetic method was also demonstrated in a continuous-flow reactor (Fig. 3B-c). At the flow rate of 0.8 mL min⁻¹, the desired 1,2-amino oxygenation was completed smoothly to produce 1.88 g products in 81% isolated yield. In addition, **4ba** and **6a** were also well performed in continuous-flow conditions toward a gram-scale synthesis of β-amino alcohol derivatives. Furthermore, this transformation was applied in the synthesis of **8**, which could further furnish a natural product, halostachine (Fig. 3B-d)[40].

## Mechanistic studies

Then, several experiments were carried out to explore the mechanism of this electro-chemical 1,2-amino oxygenation (Fig. 4A). As shown in Fig. 4A-a, the NMR experiments supported a proton transfer progress between **1a** and DBU before electro-induced progress. When sulfonamides **10** was utilized for this transformation under standard conditions (Fig. 4A-b), both desired 1,2-amino oxygenation product **10a** and a intramolecular cyclization product **10b** via Hofmann–Löffler–Freytag reaction were obtained. These results supported the involvement of NCRs progress in the reaction

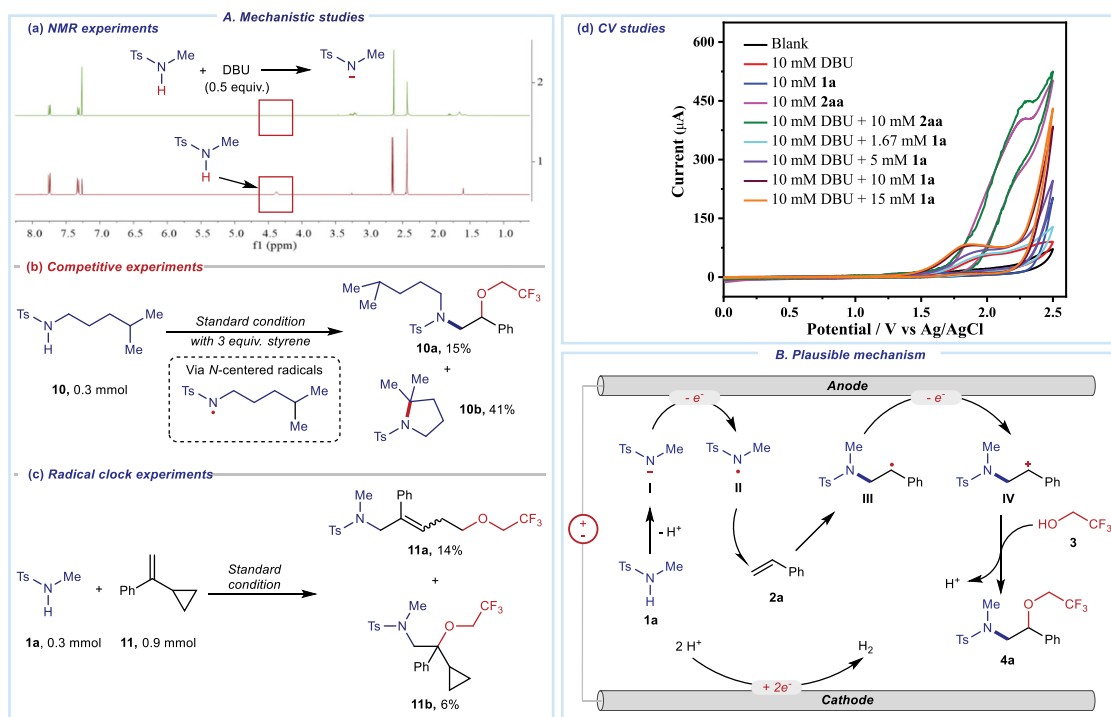

**Fig. 4 | Mechanistic studies and plausible mechanism. A** Mechanistic studies. **a** NMR experiments. **b** Competitive experiments between intermolecular 1,2-amino oxygenation and intramolecular amination. **c** Radical clock experiments. **d** CV studies. **B** Plausible mechanism.

system[37,41]. In addition, the radical clock experiment was carried out with the use of alkene **11** (Fig. 4A-c). The formation of the corresponding ring-opening adduct **11a** further supported the radical addition of in-situ formed NCRs to alkenes. Afterward, a series of CV studies were performed to explore the anodic reaction (Fig. 4A-d). Although the oxidative peak of **1a** was not observed in the range of 0–2.5 V, the catalytic current of DBU obviously increased with the addition of **1a**. These results revealed that DBU was electroactive, supporting a fast electron-transfer progress between oxidized DBU and **1a**. While styrene **2aa** showed an electro-redox activity, styrenes with electron-donating groups led to low yields (Supplementary Fig. 6). Therefore, the direct oxidation of alkenes may not be the initial step of this electro-oxidative transformation.

Based on the above results and previous studies[37,39], a plausible mechanism was proposed for this electro-oxidative 1,2-amino oxygenation (Fig. 4B). Firstly, a proton transfer between sulfonamide **1a** and DBU happened to generate N-centered anion **I**, which could be oxidized toward NCR **II** in the anode. Another mechanism could not be ruled out that a mixture of **1a** and DBU was directly oxidized in the anode to afford NCR **II**. Subsequently, such NCR **II** was added to the alkene **2a** to form a C-centered radical **III**, which could further be oxidized for the formation of C-centered cation **IV**. Then intermediate **IV** was attacked by trifluoroethanol **3** with the release of a proton. In the cathode, two protons were reduced to evolute hydrogen.

In summary, we have developed an electro-oxidative 1,2-amino oxygenation of alkenes using readily available sulfonamides and alcohols as nitrogen and oxygen sources with $H_2$ evolution. This method offers a convenient and powerful synthetic approach toward β-amino alcohols in one step without extra-oxidants. Moreover, the wide scope, good performance in continuous-flow electro-reactor, and the efficient synthesis of natural products illustrate the potential applicability of this method in the industry. Furthermore, this method can be extended to other alkene amino-functionalizations, thereby having the potential to inspire advances in other transformations via electro-induced NCRs.

## Methods

In an oven-dried undivided three-necked bottle (10 mL) equipped with a stir bar, sulfonamide substrate **1a** (0.3 mmol) and $^{n}Bu_4NBF_4$ (0.3 mmol) were combined and added. The undivided cell was equipped with graphite rod anode (φ 6 mm), platinum plate cathode (1.5 cm × 1.5 cm × 0.3 mm) and was then charged with nitrogen. Under the atmosphere of nitrogen, DBU (0.6 mmol), and alkenes **2** (0.9 mmol) were added, then DCM (4.0 mL) and TFE (2.0 mL) were injected respectively into the tubes via syringes. The mixture was electrolyzed using constant current conditions (4.0 mA) for 5 h at room temperature under magnetic stirring. When TLC analysis indicated that the electrolysis was complete (witnessed by the disappearance of the **1a**), the solvent was removed under reduced pressure. The residue was purified by column chromatography on silica gel using a mixture of PE/EA (v:v = 25:1) as eluent to afford the desired pure product.

## Data availability

The authors declare that the data supporting the findings of this study are available within the paper and its Supplementary information files or from the authors upon request.

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

## Acknowledgements

This work was supported by the National Key R&D Program of China No. 2021YFA1500100 (A.L.), National Natural Science Foundation of China 22031008 (A.L.), 21762025 (T.W.), Science Foundation of Wuhan 2020010601012192 (A.L.), and Key Projects of Natural Science Foundation of Jiangxi Province 20192ACBL20026 (T.W.). We thank W.L. (WHU) for the helpful discussions.

## Author contributions

A.L. and S.W. conceived the work. S.L., S.W., P.W., Z.H., and T.W. designed the experiments and analyzed the data. S.L. performed the experiments. S.W. described the manuscript.

## Competing interests

The authors declare no competing interests.
