## [Peer Review File · Nature Communications]

REVIEWER COMMENTS

Reviewer #1 (Remarks to the Author):

The paper by Lei and coauthors describes an electro-induced method for furnishing β -amino alcohols derivatives from alkenes under mild conditions. This work has well performed the application of NCRs in 1,2-amino oxygenation with a good regio-selectivity. I believe that the operational simplicity, efficacy and broad scope of this highly regio-selective, multicomponent electro-oxidative amino-alcohol synthesis will leverage it with widespread use in organic chemistry. I think this work is an important replenishment of electro-induced alkene di-functionalization in which the intermolecular amino-functionalization is desired, yet rarely reported. Therefore, this method also has the potential to inspire further advances in electrochemistry. Overall, I think this work is worth to be published in Nature Commun.

In the synthetic part, the authors have well demonstrated the wide scope of this method that both styrene and alkenyl ether are suitable for this transformation. In synthetic applications, the expansions of this strategy have been demonstrated fairly well. In addition, it's interesting that the yield of scale-up experiments with continuous-flow reactor is better than it with an undivided cell (in SI), thus showing the advantage of their flow reactor and the application potential of this method in further electro-industry. Is this result sustained for other substrates?

In mechanistic studies, they proposed a plausible mechanism with server experiments. Although it seems possible, the competitive experiments are not rigorous enough. The benzyl C-H bond of 10 can be oxidized to generate carbon cation and then the product 10b is formed. It is better to choose another substrate for competitive experiments.

All in all, I think this work can be published in Nature Commun. with the minor corrections detailed above.

Reviewer #2 (Remarks to the Author):

In this manuscript, Lei and coauthors report an electro-oxidative three-component approach toward β -amino alcohol derivatives from alkenes and sulfamides under mild conditions. Although electro-induced alkene 1,2-amino oxygenation has been reported by Lin (J. Am. Chem. Soc. 2018,140, 12511), this work shows a more general scope for both oxygen and nitrogen sources. This is one point to convince me that this work may be suitable for Nat. Commun. Moreover, the strategic expansion and synthetic application has well demonstrated the merit of this work and make it of general interest in areas that range from agrochemical to pharmaceutical industries. However, it seems like a drawback that the scope is limited in mono- or 1,2-disubstituted alkenes and the scope with other type of alkenes is lack of discussion. The authors designed this work based on their previous researches, thereby proving a plausible mechanism in manuscript. But the competitive experiments in Fig 2g is not powerful to support a NCRs route. Overall, I think this work is worth to be published in Nature Commun. after addressing the comments below:

1. An additional scope of alkene should be added, especially for di-, tri-, and tetra-substituted alkenes.
2. The competitive experiments should be carried out with non-aryl substrates.
3. Some complex molecules that derived from natural products or drugs should be examined. These experiments could probe whether this method is suitable for late-stage functionalization.

All in all, I think this work provides an elegant method to achieve amino oxygenation of alkenes and it can be published in Nature Commun. with the minor corrections detailed above.

Reviewer #3 (Remarks to the Author):

This is an interesting paper reporting the electrochemical oxyamination of alkenes to yield products of industrial and pharmacological interest.

The anodic oxidation of the anion of tosylamine yields the corresponding N radical which, in turn, adds to the double bond, yielding a carbon radical.

This last can be oxidised at the anode to cation and react with an oxygen source yielding the product.

The reaction scope is really broad, also if the yields are seldom high. This electrochemical reaction allows to avoid stoichiometric amount of redox reagents and the consequent formation of waste.

I think that this paper can be of interest for the readers of Nature Communications, but I have some questions.

-The authors wrote that in some cases the alkene can be oxidised in place of the tosylamine. Why did they not prove this carrying out cyclic voltammetries?

- The used an excess of base to carry out the tosylamine deprotonation, in order to have a more reactive species at the anode. The electrolysis is carried out in a undivided cell and the cathodic reaction is the hydrogen evolution with formation of alkoxide anions. Why an excess is base is bìnecessary? During electrolysis the alkoxide ion could act as a base. Have the authors tried to use lower amounts of DBU?

- I think that the voltammetric analysis does not support the hypothesis of mechanism. In particulae it seems that the tosylamine oxidises at a less positive potential that the corresponding anion (?), i.e., than the solution containing the tosylamine and DBU. In particular, it seems that only DBU is electroactive.

I think that CVs in the presence of alkene could be useful and a discussion of the CVs in SI is necessary.

Reviewer #1 (Remarks to the Author):

The paper by Lei and coauthors describes an electro-induced method for furnishing β -amino alcohols derivatives from alkenes under mild conditions. This work has well performed the application of NCRs in 1,2-amino oxygenation with a good regio-selectivity. I believe that the operational simplicity, efficacy and broad scope of this highly regio-selective, multicomponent electro-oxidative amino-alcohol synthesis will leverage it with widespread use in organic chemistry. I think this work is an important replenishment of electro-induced alkene di-functionalization in which the intermolecular amino-functionalization is desired, yet rarely reported. Therefore, this method also has the potential to inspire further advances in electrochemistry. Overall, I think this work is worth to be published in Nature Commun.

Response: Thanks to this **Reviewer** for his/her recognition of our work.

In the synthetic part, the authors have well demonstrated the wide scope of this method that both styrene and alkenyl ether are suitable for this transformation. In synthetic applications, the expansions of this strategy have been demonstrated fairly well. In addition, it's interesting that the yield of scale-up experiments with continuous-flow reactor is better than it with an undivided cell (in SI), thus showing the advantage of their flow reactor and the application potential of this method in further electro-chemistry. Is this result sustained for other substrates?

Response: We appreciate the positive point and suggestion of this **Reviewer**. For better demonstrating the merit of this electro-chemical 1,2-amino oxygenation, we carried this synthetic method with **4ba** or acid in continuous-flow conditions. We also added the related discussion in revised manuscript as follow:

“In addition, **4ba** and **6a** were also well performed in continuous-flow conditions towards a gram-scale synthesis of β -amino alcohol derivatives.”

Revised Figure 3B-III:

In mechanistic studies, they proposed a plausible mechanism with server experiments. Although it seems possible, the competitive experiments are not rigorous enough. The benzyl C-H bond of 10 can be oxidized to generate carbon cation and then the product 10b is formed. It is better to choose another substrate for competitive experiments.

Response: We fully agree the point of this **Reviewer**, and we have revised the manuscript with another substrate as follow:

All in all, I think this work can be published in Nature Commun. with the minor corrections detailed above.

Response: We acknowledge again for the detailed comments and suggestions from the first reviewer. We are sure that the quality of this work will be improved after considering these helpful suggestions.

Reviewer #2 (Remarks to the Author):

In this manuscript, Lei and coauthors report an electro-oxidative three-component approach toward β -amino alcohol derivatives from alkenes and sulfamides under mild conditions. Although electro-induced alkene 1,2-amino oxygenation has been reported by Lin (J. Am. Chem. Soc. 2018,140, 12511), this work shows a more general scope for both oxygen and nitrogen sources. This is one point to convince me that this work may be suitable for Nat. Commun. Moreover, the strategic expansion and synthetic application has well demonstrated the merit of this work and make it of general interest in areas that range from agrochemical to pharmaceutical industries. However, it seems like a drawback that the scope is limited in mono- or 1,2-disubstituted alkenes and the scope with other type of alkenes is lack of discussion. The authors designed this work based on their previous researches, thereby proving a plausible mechanism in manuscript. But the competitive experiments in Fig 2g is not powerful to support a NCRs route. Overall, I think this work is worth to be published in Nature Commun. after addressing the comments below:

Response: Thanks to this **Reviewer** for his/her positive comments.

1. An additional scope of alkene should be added, especially for di-, tri-, and tetra-substituted alkenes.

Response: Thanks for the suggestions. We have tried di-, tri-, and tetra-substituted alkenes. 1,1-Disubstituted alkene **2ba** was transformed to corresponding products in 83% yield. However, allylic amination has been achieved with tri- and tetra-substituted alkenes in this electro-chemical conditions (shown in follow figure). These results

proved a carbon cation was involved in our system, thereby supporting our proposed mechanism. More efforts on allylic amination are currently going on.

2. The competitive experiments should be carried out with non-aryl substrates.

Response: We fully agree the point of this **Reviewer**, and we have revised the manuscript with another substrate as follow:

3. Some complex molecules that divided from natural products or drugs should be examined. These experiments could probe whether this method is suitable for late-stage functionalization.

Response: We have examined several molecules that divided from natural products or drugs. However, only menthol and indanol derivatives were tolerated in standard conditions, forming desired products in low yields.

In our recently report (J. Am. Chem. Soc. 2021, 143, 49, 20863), we found that NCRs have a potential to react with C(sp³)-H bond via a HAT progress, thereby maybe making a difficulty in the LCF of this method. These discussions have been added in SI.

“To examine whether this synthetic strategy is suitable for the late-stage functionalization of complex molecules, several molecules that divided from natural products or drugs have been tested in standard conditions. However, only menthol and indanol derivatives were tolerated in standard conditions, forming desired products in low yields. In addition, molecules from diacetone-D-glucose or cholesterol were failed to afford corresponding products. Combined our previous work, these results may cause by the feature of NCRs that have a potential to react with C(sp³)-H bond via a HAT progress.”

Supplementary Figure 8. Scope of molecules that divided from natural products or drugs

All in all, I think this work provides an elegant method to achieve amino oxygenation of alkenes and it can be published in Nature Commun. with the minor corrections detailed above.

Response: We acknowledge again for the detailed comments and suggestions from the second reviewer. We are sure that the quality of this work will be improved after considering these helpful suggestions.

Reviewer #3 (Remarks to the Author):

This is an interesting paper reporting the electrochemical oxyamination of alkenes to yield products of industrial and pharmacological interest.

The anodic oxidation of the anion of tosylamine yields the corresponding N radical which, in turn, adds to the double bond, yielding a carbon radical.

This last can be oxidised at the anod to cation and react with an oxygen source yielding the product.

The reaction scope is really broad, also if the yields are seldom high. This electrochemical reaction allows to avoid stoichiometric amount of redox reagents and to the consequent formation of waste.

I think that this paper can be of interest for the readers of Nature Communications, but I have some questions.

Response: Thanks to this **Reviewer** for his/her recognition of our work.

-The authors wrote that in some cases the alkene can be oxidised in place of the tosylamine. Why did they not prove this carrying out cyclic voltammetries?

Response: Thanks for helpful suggestions. We have added CV data of several alkenes and related discussion in our SI as follow:

Supplementary Figure 6. The cyclic voltammograms in solvent (6 mL) by using glassy carbon as the working electrode, Pt wire as the counter electrode and Ag/AgCl as the reference electrode under N₂ at room temperature. The scan rate is 0.1 V/s. Black line: 4 mL DCM and 2 mL TFE, TBABF₄ (0.1 M). Red line: 10 mM 4-Methoxystyrene. The oxidation peaks of red line were observed at 1.75 V (vs Ag/AgCl). Blue line: 10 mM DBU, 15 mM 1a (*N*-Methyl-*p*-toluenesulfonamide). The oxidation peaks of blue line were observed at 1.85 V (vs Ag/AgCl).

“Alkenes with electron-donating groups showed lower oxidation potentials than the mixture of DBU and amide **1a**, proving a prioritized oxidation of alkenes in the anode. Therefore, with these alkenes, the formation of NCRs may be suppressed, causing the low yields in this electro-chemical oxygen amination (Supplementary Figure 6).”

- The used an excess of base to carry out the tosylamine deprotonation, in order to have a more reactive species at the anode. The electrolysis is carried out in a undivided cell and the cathodic reaction is the hydrogen evolution with formation of alkoxide anions. Why an excess is base is bivnecessary? During electrolysis the alkoxide ion could act as a base. Have the authors tried to use lower amounts of DBU?

Response: Thanks very much for the reviewer’s kind suggestions. We have explored the impact of yields and DBU equivalents. As shown in the follow table, 2 equiv. DBU could promoted the desired transformation better than 1 equiv. DBU.

Entry	Equivalent of DBU	Isolated yield of 4aa
1	0	N.d.
2	0.5	12%
3	1.0	41%
4	1.5	64%
5	2.0	76%

We have carried several experiments to investigate the effects of DBU. Since TFE is one type of weak acid (pK_a = 12.4 in 25 °C), excessive DBU may play a role of base to increase the concentration of N-centered anions.

Moreover, according to the CV results (Figure 4-IV and Supplementary Figure 7), DBU is electroactive and the current is increased with the addition of amines. These results support that excessive DBU may promote the electron transform of amides in the electrode surface.

- I think that the voltammetric analysis does not support the hypothesis of mechanism. In particular it seems that the tosylamine oxidises at a less positive potential than the corresponding anion (?), i.e., than the solution containing the tosylamine and DBU. In particular, it seems that only DBU is electroactive.

Response: Thanks very much for the reviewer's constructive comments. We have added detail CV data and related discussion in SI.

“ Afterwards, a series of CV studies were performed to explore the anodic reaction (Figure 4A-IV). Although the oxidative peak of **1a** was not observed in the range of 0 – 2.5 V, the catalytic current of DBU obviously increased with the addition of **1a**. These results revealed that DBU was electroactive, supporting a fast electron-transfer progress between oxidized DBU and **1a**. While styrene **2aa** showed an electro-redox activity, styrenes with electron-donating groups led to low yields (Supplementary Figure 6). Therefore, the direct oxidation of alkenes may not be the initial step of this electro-oxidative transformation.”

I think that CVs in the presence of alkene could be useful and a discussion of the

CVs in SI is necessary.

Response: We acknowledge again for the detailed comments and suggestions from the third reviewer. The CV data and related discussion have been well supported in the revised manuscript.

REVIEWERS' COMMENTS

Reviewer #1 (Remarks to the Author):

My concerns have been addressed in the revisions. I think this work can be published in Nature Communication.

Reviewer #2

< In comments to the editorial office, this reviewer expressed that the revised manuscript is ready for publication. >

Reviewer #3 (Remarks to the Author):

The revised version of the paper is suitable for publication in Nature Communications

Reviewer #1 (Remarks to the Author):

My concerns have been addressed in the revisions. I think this work can be published in Nature Communication.

Response: Thanks to this *Reviewer* for his/her recognition of our work. We acknowledge again for the detailed comments and suggestions from the first reviewer.

Reviewer #2 (Remarks to the Author):

< In comments to the editorial office, this reviewer expressed that the revised manuscript is ready for publication. >

Response: Thanks to this *Reviewer* for his/her recognition of our work. We acknowledge again for the detailed comments and suggestions from the second reviewer.

Reviewer #3 (Remarks to the Author):

The revised version of the paper is suitable for publication in Nature Communications

Response: Thanks to this *Reviewer* for his/her recognition of our work. We also appreciate these valuable comments from all referees who help us a lot to improve our understanding and the quality of our manuscript.